# Lateral Peri-Implantitis: Successful Management via Guided Bone Regeneration at Mandibular First Molar Implant

**DOI:** 10.3390/medicina59091691

**Published:** 2023-09-21

**Authors:** Won-Bae Park, Michael Villa, Ji-Young Han, Hyun-Chang Lim, Philip Kang

**Affiliations:** 1Department of Periodontology, School of Dentistry, Kyung Hee University, Seoul 02447, Republic of Korea; wbpdds@naver.com; 2Private Practice in Periodontics and Implant Dentistry, Seoul 02771, Republic of Korea; 3Division of Periodontics, Section of Oral, Diagnostic and Rehabilitation Sciences, College of Dental Medicine, Columbia University, New York, NY 10032, USA; mdv2128@cumc.columbia.edu; 4Department of Periodontology, Division of Dentistry, College of Medicine, Hanyang University, 222-1 Wangsimni-ro, Seongdong, Seoul 04763, Republic of Korea; hjyperio@hanyang.ac.kr; 5Department of Periodontology, Periodontal-Implant Clinical Research Institute, School of Dentistry, Kyunghee daero 23, Dongdaemoon, Seoul 02447, Republic of Korea

**Keywords:** complication, dental implant, guided bone regeneration, peri-implantitis

## Abstract

Infections occurring around implants are divided into marginal peri-implantitis and retrograde peri-implantitis (RPI). Marginal peri-implantitis starts in the crestal bone and progresses to the apical portion, and RPI starts in the apical bone and progresses to the coronal portion. However, lateral peri-implantitis (LPI) occurring on the side of the implant body has not yet been reported, and the cause is unclear. This 63-year-old male patient is a case of unusual bone resorption that occurred in the lateral portion of the implant body 26 months after lateral bone augmentation. The origin of LPI was an infection at the site of laterally augmented bone. Rather than implant removal, this report demonstrates an alternative treatment option of guided bone regeneration after the enucleation and detoxification of the implant surface with successful clinical and radiographic results for 2 years.

## 1. Introduction

Peri-implantitis is described as a pathological condition, wherein the affected dental implant is surrounded by inflamed connective tissue and a progressive loss of supporting bone [1]. Evidence from human biopsies demonstrates a histological infiltrate of inflammatory cells in the soft tissue surrounding implants challenged by plaque formation [2]. The consistent plaque challenge and inflammatory process in ligature models have demonstrated a conversion of healthy implants to peri-implantitis with severe inflammation and accompanied tissue destruction [3]. 

The most commonly observed peri-implantitis cases can be attributed to the host response to biofilm formation [4]. The diseased implant is accompanied by a morphological osseous defect typically characterized by infra-osseous defect or a supra-crestal horizontal defect or a combined defect [1]. The osseous defects shows a corono-apical trend, and the prevalence has heterogenicity, with the ratio being about 20% [5,6]. In contrast, retrograde peri-implantitis (RPI), identified at the apex of the implant, with an apico-coronal trend of the osseous defect is relatively rare [7,8].

Retrograde peri-implantitis (RPI) is defined as a clinically symptomatic peri-apical lesion that develops shortly after implant insertion, while the coronal portion of the implant achieves a normal bone to implant interface [8]. The diagnosis of RPI is generally conducted through clinical and radiographic findings and can be confirmed within the first 8 weeks of implant placement [9,10,11,12]. However, cases with no or weak clinical symptoms are sometimes detected late [9]. Surgical intervention is required when clinical symptoms are present or the size of the periapical radiolucent lesion gradually increases [13]. First identified by McAllister et al. [14], RPI was associated either with previous periapical lesions in the extracted teeth or neighboring teeth at the infected implant. The causes of RPI were than attributed to be endodontic in nature or apical bone necrosis due to over-heating during implant site preparation, cortical bone perforation and implant body contamination [15].

Studies on RPI are mainly case reports or case series [10,16,17,18], and a small number of retrospective analyses [12] with a review [9]. The purpose of this case report is to introduce the first case of lateral peri-implantitis (LPI) occurring adjacent to the lateral bone augmentation site which had received implant surgery 3 years prior and was successfully treated for LPI. We define lateral peri-implantitis as an inflammatory radiolucency observed at the lateral surface of the osseous contact to the implant body where the apical and coronal osseous crest is unaffected. 

## 2. Case Presentation

A 63-year-old male patient with no systemic health conditions and no history of smoking presented to the clinic with hypermobility of the mandibular right first molar. Panoramic radiographs and cone beam computed tomography (CBCT) were acquired and interpreted. Severe bone resorption was observed in the residual distal root of the first molar (Figure 1a). The tooth previously received a mesial root amputation. Three years post-amputation, saving the tooth was deemed hopeless, and it was subsequently extracted. Four months post-extraction, the healed socket site showed an atrophied soft tissue contour. An implant placement was planned to replace the missing tooth.

### 2.1. Implant Placement and Lateral Bone Augmentation

Under local anesthesia with lidocaine (containing 1:100,000 epinephrine), mid-crestal and vertical incisions were performed. Using the periosteal elevator, the mucoperiosteal flaps were reflected buccally and lingually. The site was degranulated thoroughly. An osteotomy was performed under saline irrigation, where the implant site was prepared according to the surgical specification of the implant manufacturer to accommodate a size 4.3 × 12 mm Implantium implant (Dentium, Suwon, Republic of Korea). The implant placed was submerged with a cover screw, and an Osteon III bone graft (biphasic calcium phosphate, Genoss, Suwon, Republic of Korea) was placed at the buccal site followed by a resorbable collagen membrane (Genoss, Suwon, Republic of Korea). The collagen membrane was trimmed to be slightly smaller than the defect size and placed with no contact to the incision line. The mucoperiosteal flap was closed without tension using 4-0 nylon sutures (Ethilon^®^ 4.0, Ethicon, Cincinnati, OH, USA). Systemic antibiotics (Cefradine 500 mg, Yuhan Pharmaceutical Co., Ltd., Seoul, Republic of Korea) and a nonsteroidal anti-inflammatory drug (Etodol^®^ 200 mg, Yuhan Co., Seoul, Republic of Korea) were administered three times a day for 10 days. Patients were advised to rinse with 0.12% chlorhexidine solution (Hexamedine, Bukwang Pharmaceutical, Seoul, Republic of Korea) for 2 weeks. The sutures were removed 10 days post-surgery.

### 2.2. Uncovering Procedure

The healing was uneventful; at 5 months, second stage surgery was performed to uncover the implant. After local anesthesia, the buccal flap was reflected. The cover screw was located and replaced with a healing abutment. The unabsorbed part of the collagen membrane was removed. The grafted buccal site healed well, and a localized area on mesial portion of the graft was observed to have a yellowish hue (indicated by the arrow in Figure 2b). Bone sounding with the periodontal probing of the grafted site demonstrated that the mesial portion was more easily perforated than the distal portion. The flaps were closed with 4-0 nylon sutures. A systemic antibiotic (Cefradine 500 mg, Yuhan Pharmaceutical Co., Ltd., Seoul, Republic of Korea) and a non-steroidal anti-inflammatory drug (Etodol^®^ 200 mg, Yuhan Co., Seoul, Republic of Korea) were administered three times a day for 5 days. The sutures were removed after 10 days.

### 2.3. Prosthesis Delivery and Follow-Up 

Four weeks post-uncovering, an impression was acquired and subsequently the final prosthesis was delivered. Panoramic radiographs were taken, and no abnormalities were found. No specific clinical symptoms were documented in the follow ups. However, 26 months after lateral augmentation (19 months after prosthesis installation), the patient visited the clinic due to edema and the bleeding of the buccal gingiva (Figure 3a). Panoramic radiographs and CBCT scans were taken. 

### 2.4. Surgical Reentry

The buccal mucoperiosteal flap was reflected under local anesthesia. The soft tissue inside the bony defect around the implant was observed to be connected to the flap’s inner surface, and the area was carefully dissected. The granulation tissue at the bone defect was debrided and detached from the implant body and bone surface using hand instruments. The inner portion of the granulation tissue was in contact with the implant body, and the outer portion was the grafted site where bone grafting was performed. The removed specimen was fixed in 10% formalin for histopathological examination. The crestal bone at the implant–abutment connection was observed to be intact, and the implant was observed to be osseointegrated; therefore, implant removal was ruled out. The exposed implant surface was treated with a titanium brush to remove any debris and to obtain a polished appearance following the manufacturer’s guidelines. A 5 min surface application with 100 mg/mL tetracycline HCl (tetracycline HCl 500 mg powder mixed with 5 cc physiological saline solution) mixed with saline was performed for additional decontamination. Sufficient saline irrigation was carried out. The bone defect was filled with Osteon III (Genos, Suwon, Republic of Korea), covered with a resorbable collagen membrane (Genoss, Suwon, Republic of Korea), and then sutured (Figure 3f–h). Antibiotics and analgesic anti-inflammatory drugs were prescribed for 10 days. The suture was removed after 10 days.

### 2.5. Histopathological Examination

The removed specimen was about 13 × 11 mm in size. Specimens fixed in 10% neutral buffered formalin were divided into two sections. One portion of the biopsy was decalcified. The sample that did not go through decalcification, the specimen was embedded with resin. The decalcified specimen was embedded in a paraffin block and cut into 4 um using a microtome. The embedded resin was cut with 50 μm thickness using a grinder and was stained with hematoxylin and eosin (Figure 4). The image was taken using a digital slide scanner (PANNORAMIC 250 Flash III, 3DHISTECH Ltd., Hungary) and observed with a Caseviewer (3DHISTECH Ltd., H-1141 Budapest, Hungary).

### 2.6. Radiographic Evaluation

Panoramic radiography (Figure 5a–c) and CBCT (Figure 5d,e) images were taken during the course of treatment. In the radiographs taken after the implant prosthesis was installed, no abnormal images were found (Figure 5a). However, 26 months after the prosthesis was installed, an ovoid-shape bone resorption was observed around the implant body (Figure 5b). Twenty months after surgical intervention, including GBR, no bone resorption around the implant was observed (Figure 5c). On the coronal image of the CBCT taken just before the surgical intervention, extensive bone resorption and perforation of the cortical layer were observed on the buccal side (Figure 5d). On the CBCT image taken 20 months after surgical intervention, good bone regeneration was observed (Figure 5e).

## 3. Result

The presented case is the first known clinical and radiographic presentation of bone resorption localized laterally to implant with intact coronal and apical bone to implant contact. The processed histological samples illustrate a chronic inflammatory response to the bone graft (Figure 4b–e). Inflammatory cells surround the soft tissue, specifically osteoclast cells along with giant body cells are noted surrounding the graft. At second stage, the graft material at the mesiobuccal aspect appeared to be soft in nature. Post-final restoration of the implant crown, several months later, the mesiobuccal site displayed bone resorption extending laterally. We therefore hypothesize that the graft material could be the source of infection triggering a cascade of inflammatory response leading to bone resorption. Bone resorption noted on the lateral buccal side of the implant was exacerbated by the active infection of laterally augmented graft material. One month after the treatment of LPI using surface decontamination and regrafting, the patient’s clinical symptoms disappeared and no inflammatory findings occurred at the surgical site, so the existing prosthesis was re-installed. After 20 months, no clinical and radiological problems were observed, and bone formation around the bone defect was confirmed. 

## 4. Discussion

In the present case, although the cause of LPI is unclear, it has been suggested that infection from the laterally augmented site was transmitted to the implant body. This was confirmed by histological findings. The treatment of LPI was GBR using a bone graft and resorbable collagen membrane after surgical enucleation and surface decontamination, and effective results were achieved. 

Although RPI has been reported to be detected between 1 week and 4 years, it is generally found within 6 months of implant placement [9]. The present case was confirmed radiographically with the onset of clinical symptoms 26 months after lateral bone augmentation. The actual bone resorption process seems to have started to occur earlier than this. Various factors were introduced as causes of RPI, but most were endodontic infection from adjacent teeth [9]. The reason for this is that while the prevalence of RPI is very low at 0.26–1.86% [12], the implant near the endodontic periapical lesion showed a high prevalence of about 7.8% [18]. This is believed to be caused by bacteria in the apical lesion of adjacent teeth activated during implant drilling [12]. However, in the present case, 3 years have passed since the extraction of the mesial root of tooth #30. Therefore, the possibility of the bacterial activation of the endodontic periapical lesion is very low. 

The tissue regenerated in the GBR procedure exhibits various characteristics from bone tissue to inflamed granulation tissue. Causes of adverse healing include flap or bone damage during surgery, contamination during delivery of bone graft substitute, influx of bacteria due to the exposure of wound edges, and delayed side effects due to incorrect postoperative management [19]. During uncovering in the present case, the mesial portion of the laterally augmented graft showed a yellowish color compared to the distal portion and was easily penetrated during periodontal probing. However, we did not remove it, believing that it would consolidate over time. However, the augmented bone of this mesial portion eventually became infected. The influx of contaminants to the lateral augmented site appears to have been through the gap between the membrane and the bone graft substitute. 

There is still no standardized consensus on the treatment of RPI [15]. Therapeutic modalities of RPI include the administration of antibiotics [20], degranulation with/without apical implant resection [8,11,21], implant surface treatment with/without chemical detoxification [8], and bone grafts with/without a barrier membrane [3,11,16]. In addition, there is heterogenicity regarding the operator’s the choice of treatment and the treatment results. The implant survival rate after treatment also varied from 67.5% to 97.4%, with some cases showing 100% survivability [12,15,20].

The treatment of LPI does not appear to be much different from the treatment of RPI. However, the cause and location of infection are different, and implant apex resection is not required. In the present case, after separating and removing the inflammatory granulation tissue attached between the buccal mucosa and the implant, the contaminated implant body was mechanically decontaminated. However, there is no standardized method for implant decontamination. It is common to perform chemical decontamination after mechanical decontamination. As chemical agents, tetracycline HCl, citric acid, chlorhexidine, and calcium hydroxide have been reported [17,22,23]. In the present case, detoxification was performed with tetracycline HCl for about 5 min after using the titanium brush. The defect was sufficiently irrigated with saline, and a bone graft with resorbable collagen membrane was performed. This treatment protocol has been applied to an implant apex exposed in the postoperative maxillary cyst [24] and maxillary sinus with sinus pathologies [25], and successful clinical results have been achieved.

After treatment, clinical symptoms disappeared, radiopacity increased on radiographs, and no recurrence was found. The case report presented here is the first recorded case, to the authors’ knowledge. Further investigation into similar clinical presentations are warranted. 

## 5. Conclusions

Within the limitation of this case report, the infection of the lateral augmented bone can cause LPI. However, LPI can be successfully treated with lesion enucleation, surface decontamination, and GBR techniques without implant removal.

## Figures and Tables

**Figure 1 medicina-59-01691-f001:**
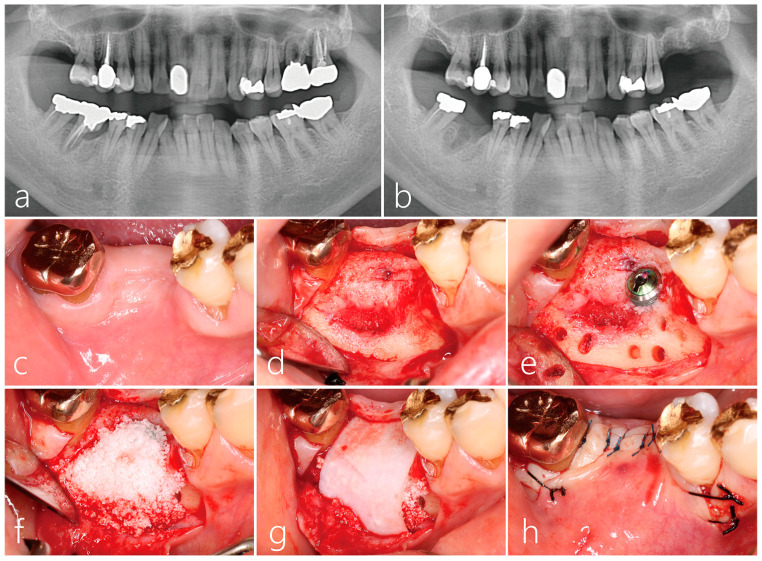
(**a**) A periapical lesion on the distal root of tooth #30 was observed. The mesial root was removed 3 years ago; (**b**) panoramic radiograph after 4 months of tooth extraction. A wide bone defect was observed in the extraction socket; (**c**) clinical findings 2 months post-extraction. The depression of the buccal bone was severe; (**d**) the mucoperiosteal flap was reflected and the granulation tissue in the extraction socket was thoroughly removed; (**e**) a 4.3 × 12 mm implant was placed. Bone decortication was performed on the buccal bone defect using a surgical round bur. Four osteotomy holes were formed on the mesial side and two holes on the distal side; (**f**) bone graft was performed using Osteon III; (**g**) the bone-grafted site was covered with a resorbable collagen membrane; (**h**) the flap was closed without tension.

**Figure 2 medicina-59-01691-f002:**
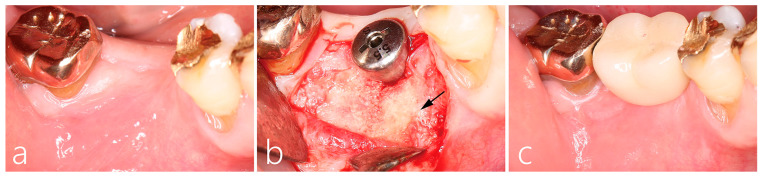
(**a**) Clinical findings of GBR post-5 months. The GBR site was not exposed; (**b**) uncovering was performed 5 months post-GBR procedure. Bone regeneration was well performed in the crestal portion of the implant–abutment junction. Of the buccal labial augmentation sites, only the mesial side showed a yellowish hue, and bone sounding with a periodontal probe presented with a soft consistency (black arrow); (**c**) the final prosthesis was inserted 1 month post-uncovering.

**Figure 3 medicina-59-01691-f003:**
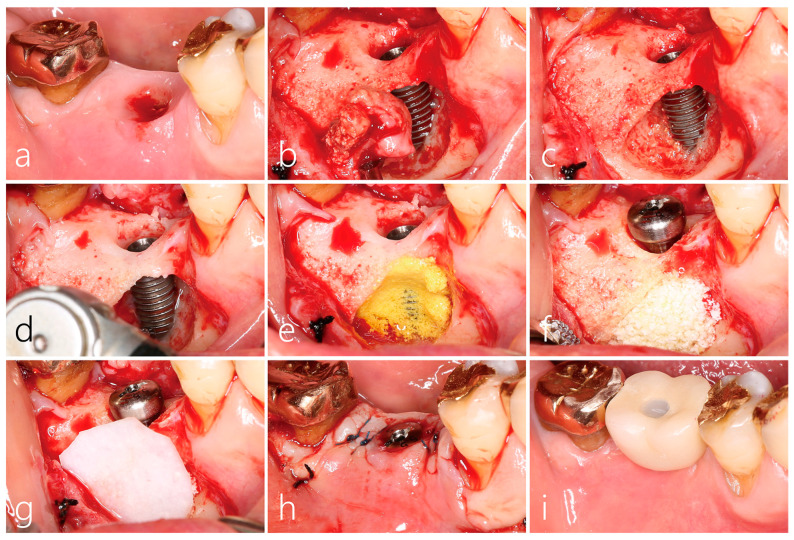
(**a**) There is gingival bleeding and swelling present 26 months after lateral bone augmentation. The prosthesis was disconnected by removing the internal screw through a screw channel located on the occlusal table; (**b**) after buccal flap reflection, the granulation tissue extended from the GBR site and penetrated up to the implant surface; (**c**) bone regeneration at the implant–abutment junction was well performed. The buccal implant body was exposed. Peri-implant bone defect occurred on the mesial side; (**d**) decontamination was performed on the exposed implant body using a titanium brush; (**e**) tetracycline HCl was mixed with saline and applied to the exposed implant surface; (**f**) after sufficient saline irrigation, a synthetic bone graft substitute was filled; (**g**) a resorbable collagen membrane was adapted to cover the bone graft site; (**h**) flap closure; (**i**) the previous prosthesis was reinserted following 1 month.

**Figure 4 medicina-59-01691-f004:**
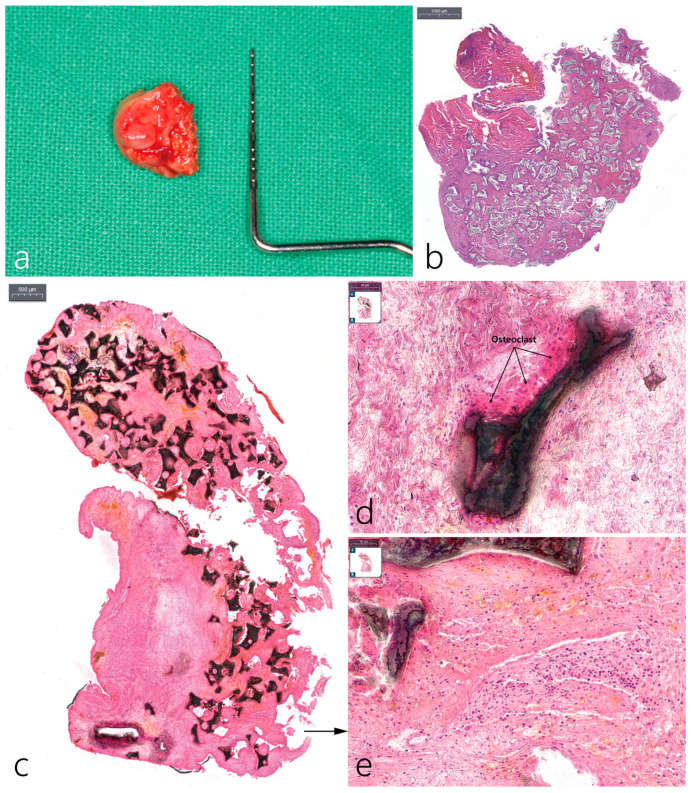
(**a**) Removed specimen measuring 13 × 11 mm. The left side is the area in contact with the implant and the right side is the area subjected to bone decortication; (**b**) in the decalcified H–E stained specimen, the area in contact with the implant is the granulation tissue containing a large amount of inflammatory cells; (**c**) in the undecalcified H–E stain specimen, graft particles did not contact the implant body; (**d**) osteoclasts appeared around the graft particle (black arrow); (**e**) abundant inflammatory cells in the granulation tissue.

**Figure 5 medicina-59-01691-f005:**
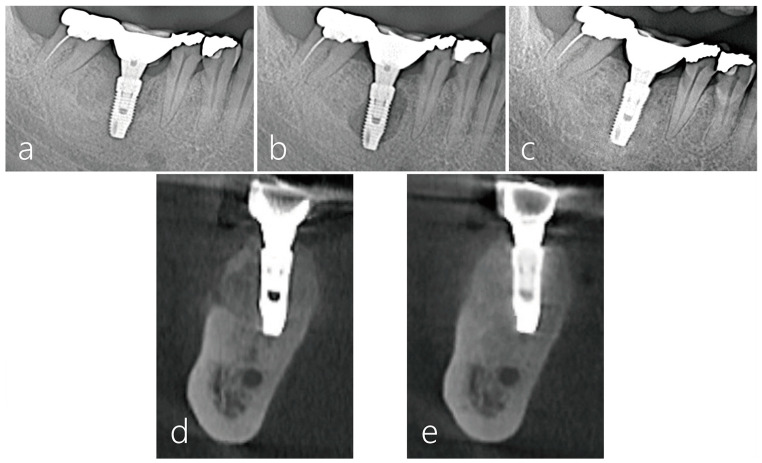
Panoramic radiograph lower right first molar region and CBCT images of the procedure: (**a**) the prosthesis was delivered 6 months post-implant placement; (**b**) 26 months after prosthesis delivery, an ovoid-type bone resorption was observed around the implant; (**c**) panoramic radiograph images taken 20 months after surgical re-entry showed no bone resorption around the implant; (**d**) in the cross-sectional image of the CBCT scan taken before the surgical re-entry, broad bone resorption was observed on the buccal side of the implant; (**e**) there was no evidence of bone resorption around the implant in the cross-sectional image of the CBCT scan 20 months after surgical re-entry.

## Data Availability

Not applicable.

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
