# Peer review of "Lateral Peri-Implantitis: Successful Management via Guided Bone Regeneration at Mandibular First Molar Implant"

_medicina, 2023, doi:10.3390/medicina59091691_

Round 1
Reviewer 1 Report
This is a well-written article. The illustrations are professional-looking and explain very well the work.
My comments:
1. Since you asked for a CBCT for the patient in preoperative, why not show the CBCT image instead of showing the panoramic X-ray only?
2. Many times during the course of the treatment and follow-up, you asked for both panoramic X-ray and CBCT together; was it not better to avoid exposure to radiation if you went for CBCT alone (ALARA principles)?
3. "45 references" for a case report is too much, I think. Try reducing the number of references, especially by removing the old ones.
Author Response
- Since you asked for a CBCT for the patient in preoperative, why not show the CBCT image instead of showing the panoramic X-ray only?
To assess the ridge thickness, authors believed that clinical photos provided enough pre-operative information, but cross-sectional views following the augmentation are included.
- Many times during the course of the treatment and follow-up, you asked for both panoramic X-ray and CBCT together; was it not better to avoid exposure to radiation if you went for CBCT alone (ALARA principles)?
You are correct. Only for the presented case, the patient was fully informed of the amount of radiation, and consent was obtained. This is not our routine practice.
- "45 references" for a case report is too much, I think. Try reducing the number of references, especially by removing the old ones.
The reference list has been shortened and the sequence has been modified accordingly.
Reviewer 2 Report
very interesting article, describing how to get out of a difficult situation during implant logical treatment and bone loss. The analysis of the collected biological material was carried out. You put forward a hypothesis about the cause of this phenomenon and tried to answer for this, good I like it!. However, as a reviewer, I have a few minor comments on your text:
Abstract
It would be good to submit the application to the Abstract. What significance does the description of your case have for others, e.g. starting with: Based on the above clinical procedure, in the case of bone loss around the implant, it is possible to consider alternative treatment consisting in......
Dentists often look for information on how to proceed in a specific case by entering a specific phrase in the Google search engine. So he will find your article and quote it.
Introduction
A few more words can be added about the treatment of this type of cases and how important prevention is.
Verardi S, Valente NA. Peri-Implantitis: Application of a Protocol for the Regeneration of Deep Osseous Defects. A Retrospective Case Series. Int J Environ Res Public Health. 2021 Dec 1;18(23):12658. doi: 10.3390/ijerph182312658. PMID: 34886384; PMCID: PMC8656633.
Raszewski Z, Nowakowska-Toporowska A, Weżgowiec J, Nowakowska D. Design and characteristics of new experimental chlorhexidine dental gels with anti-staining properties. Adv Clin Exp Med. 2019 Jul;28(7):885-890. doi: 10.17219/acem/94152. PMID: 30888120.
Case Presentation
It's a cosmetic note. Figure 1 should be placed under the text that mentions it.
Line 89
Using the periosteal elevator- type, producer country, please add for all the instrument that you are using in this case.
Line 92
prepared according to the surgical specification of the implant manufacturer,- you can add reference for this treatment protocol, form website of manufacture if it available?
Line 156
The granulation tissue at the bone defect was detached from the implant body and bone surface- can you describe it in more details, please because it is very important for the whole essence of your treatment, thank you
line
The exposed implant surface was treated with a titanium brush- producer country, rotation speed?
Discussion
Well done, I like it.
Reference
also cosmetic, but for example in the first point you don't add commas or semicolons after the author's name and surname, but in ref 4 and it's there. You have to standardize it.
good luck with your further work
Author Response
Reviewer #2:
It would be good to submit the application to the Abstract. What significance does the description of your case have for others, e.g. starting with: Based on the above clinical procedure, in the case of bone loss around the implant, it is possible to consider alternative treatment consisting in......
Dentists often look for information on how to proceed in a specific case by entering a specific phrase in the Google search engine. So he will find your article and quote it.
Added to the Abstract as recommended.
Introduction
A few more words can be added about the treatment of this type of cases and how important prevention is.
Verardi S, Valente NA. Peri-Implantitis: Application of a Protocol for the Regeneration of Deep Osseous Defects. A Retrospective Case Series. Int J Environ Res Public Health. 2021 Dec 1;18(23):12658. doi: 10.3390/ijerph182312658. PMID: 34886384; PMCID: PMC8656633.
Raszewski Z, Nowakowska-Toporowska A, Weżgowiec J, Nowakowska D. Design and characteristics of new experimental chlorhexidine dental gels with anti-staining properties. Adv Clin Exp Med. 2019 Jul;28(7):885-890. doi: 10.17219/acem/94152. PMID: 30888120.
Yes, the authors do agree with the importance of prevention. However, as the cause of retrograde peri-implantitis is still unknown, we can only treat it once it occurs.
Case Presentation
It's a cosmetic note. Figure 1 should be placed under the text that mentions it.
Once the text is finalized, figures will be re-arranged.
Using the periosteal elevator- type, producer country, please add for all the instrument that you are using in this case.
For commonly used surgical instruments like periosteal elevators, specific information about the manufacturer and origin is not required to be added to the text.
prepared according to the surgical specification of the implant manufacturer,- you can add reference for this treatment protocol, form website of manufacture if it available?
Manufacturer-specific drilling protocols may not be indicated as the objective of the case report is not to demonstrate or promote a particular product.
The granulation tissue at the bone defect was detached from the implant body and bone surface- can you describe it in more details, please because it is very important for the whole essence of your treatment, thank you
The text has been modified as suggested.
The exposed implant surface was treated with a titanium brush- producer country, rotation speed?
The text has been modified as suggested.
Reference
also cosmetic, but for example in the first point you don't add commas or semicolons after the author's name and surname, but in ref 4 and it's there. You have to standardize it.
Corrected.